# Parental perspectives on emergency health service use during the first wave of the COVID-19 pandemic in the United Kingdom: A qualitative study

Matthew Breckons[1,2]*, Sophie Thorne[3], Rebecca Walsh[4], Sunil Bhopal[1,2,5], Stephen Owens[1,5], Judith Rankin[1,2]

1 Population Health Sciences Institute, Newcastle University, Newcastle-upon-Tyne, United Kingdom, 2 NIHR Applied Research Collaboration North East & North Cumbria, Newcastle-upon-Tyne, United Kingdom, 3 Leeds Teaching Hospitals NHS Trust, Leeds, United Kingdom, 4 Imperial College Healthcare NHS Trust, London, United Kingdom, 5 Great North Children's Hospital, Newcastle upon Tyne, United Kingdom

* Matthew.Breckons@newcastle.ac.uk

**Data Availability Statement:** Our data contain potentially sensitive information and restrictions to sharing are in place based on our ethical approvals.

## Abstract

UK 'Lockdown' measures introduced in March 2020 aimed to mitigate the spread of COVID-19. Although seeking healthcare was still permitted within restrictions, paediatric emergency department attendances reduced dramatically and led to concern over risks caused by delayed presentation. Our aim was to gain insight into healthcare decisions faced by parents during the first wave of the COVID-19 pandemic and to understand if use of urgent healthcare, self-care, and information needs differed during lockdown as well as how parents perceived risks of COVID-19. We undertook qualitative telephone interviews with a purposive sample of parents living in the North East of England recruited through online advertising. We used a semi-structured interview schedule to explore past and current healthcare use, perceptions of risk and the impact of the pandemic on healthcare decisions. Interviews were transcribed and analysed using Thematic Analysis. Three major themes were identified which concerned (i) how parents made sense of risks posed to, and by their children, (ii) understanding information regarding health services and (iii) attempting to make the right decision. These themes contribute to the understanding of the initial impact of COVID-19 and associated restrictions on parental decisions about urgent healthcare for children. These findings are important to consider when planning for potential future public health emergencies but also in the wider context of encouraging appropriate use of urgent healthcare.

## Introduction

COVID-19 cases were first identified in the UK in January 2020 after which they increased exponentially. This was followed by a series of government control actions aimed at mitigating

However, data for this study are available upon reasonable request. The data request can be sent to The Faculty of Medical Sciences Research Ethics Committee, Newcastle University, UK; Email address: fmsethics@ncl.ac.uk.

**Funding:** The author(s) received no specific funding for this work.

**Competing interests:** The authors have declared that no competing interests exist.

the spread of the virus which included the closure of non-essential services, followed shortly by a national "lockdown" [1].

These measures including stringent and legally-enforceable social-distancing rules, played an important role in controlling COVID-19 in the UK from 23 March 2020 when the core government message was to "Stay home, Protect the NHS, Save Lives" [2, 3]. This messaging varied throughout the trajectory of the pandemic as lockdown restrictions were eased and tightened in response to infection rates.

While seeking healthcare was always exempt from these control measures, in the weeks that followed the first national lockdown, there was a sudden and substantial reduction in paediatric emergency department attendances [4]. Hypothesised explanations included: 1) reduced burden of infectious disease—perhaps due to reduced social mixing, 2) fear of accessing hospital, and 3) increased self-care at home—perhaps due to reduced school and workplace attendance by children and parents [5, 6]. Other suggestions included reduction in attendances for complex long-term conditions might indicate that avoidance by parents was a contributing factor [7].

In April 2020, the Royal College of Paediatrics & Child Health expressed concern over reports of delayed hospital attendance leading to a small number of children becoming unwell or dying [8]. A survey of 4,075 UK Paediatricians (>90% of all UK Paediatricians) reported nine deaths with delay in presentation as a possible contributing factor. Around a third of Paediatricians working in Emergency Departments (ED)/Paediatric Assessment Units reported witnessing at least one delayed-presentation [8].

An existing body of pre-COVID-19 literature has examined how parents make decisions about children's healthcare. A 2020 systematic review of research examining parental/family decision making regarding emergency healthcare has suggested several factors may play a role: while a large part of decision making may be perceptions of urgency, other factors such as access and positive relationships in primary care, perceived quality and speed of care and the need for reassurance also featured in decision making processes [9]. This review only spanned an early portion of the COVID-19 timeline and as such, findings regarding attendance was limited to quantitative descriptions of dramatic reductions in paediatric attendances. A more recently published qualitative study suggested that the decision to seek unscheduled care was related to parents' perceived capacity to self-manage being exceeded and that the services accessed were related to availability of General Practitioners [10]. although data was not collected past March 2020 so did not include healthcare decisions during the pandemic.

We set out to understand parents' and views on the use of children's urgent healthcare services during the first wave of the COVID-19 pandemic and in particular factors which may affect seeking care during 'lockdown'.

## Methods

We adopted a pragmatic qualitative study design as we sought to understand an ongoing real-world issue and we placed an emphasis on 'actionable learning' [11].

Ethical approval was granted by Newcastle University's Faculty of Medical Sciences Research Ethics Committee (Ref: 1919/2812).

### Participants and recruitment

Given the pace of change of the pandemic and associated restrictions, we aimed to collect data rapidly. Accessing views of people who are *not* using a service is, by its nature, challenging; we therefore sought to gain views from a more general sample of parents on using urgent healthcare during the pandemic, being aware that this sample was likely to include people who had and had not needed to make urgent healthcare decisions during the pandemic. From 22nd

May 2020, we recruited parents of children aged 0–10 years living in the North-East of England using a written advertisement containing a web link circulated via social media, charity partners, online forums and public involvement groups. On clicking this link, participants were asked to complete a consent form and a short questionnaire including demographics, previous use of children's emergency services and willingness to use these during the pandemic, willingness to participate in an interview, and contact details. We purposively sampled based on responses to obtain a maximum variation sample. Potential participants were contacted by phone or e-mail. The study rationale was discussed, any questions answered and an electronic information sheet was e-mailed.

Digitally recorded telephone interviews were conducted by male and female researchers; a postdoctoral qualitative researcher (MB), a Foundation Interim Year 1 (RW) and an Intercalating Medical Student (ST). RW and ST received supervision throughout the process. Interviewers introduced themselves as members of the research team. Due to the closure of university buildings and National regulations permitting leaving home for limited reasons during the data collection period, it was not possible to obtain hard-copies of written consent and verbal consent was audio recorded instead. We used a semi-structured interview schedule including open-ended questions exploring past and current use of children's urgent healthcare and impact of the pandemic on views on making healthcare decisions. The interview schedule was reviewed regularly to add questions and prompts to explore emerging areas of interest in subsequent interviews. Recordings were transcribed verbatim by an external transcription company and anonymised to ensure confidentiality. Researchers made field notes after each interview summarising main points, reflections and considerations for interview schedule refinement.

### Data analysis

Data were analysed according to principles of thematic analysis [12]. Analysis took place concurrently with data collection with all team members involved. Regular team meetings were held during the data collection period to review the interview schedule, interview process and aid collective familiarisation. Researcher fieldnotes fed into these discussions. A provisional coding framework, using an inductive approach to coding, was developed at whole team read-throughs of two transcripts. MB, SB, ST and RW then individually applied this to two further transcripts each before refining, agreeing a final framework and producing a codebook which included a brief description of each code. This codebook was used to code all transcripts using NVivo 12. Coded data were reviewed by all team members individually and discussed before final themes and subthemes were agreed. The decision to cease recruitment was made collectively by the research team when it was considered that saturation had been reached in terms of additional interviews not requiring additional codes [13] or leading to any new themes [14].

## Results

### Sample

We received survey responses from 121 parents. We sampled purposively, based on respondents' survey data regarding: age, sex, location, IMD decile, number and age of children, stated willingness to use health services during the pandemic and use of healthcare during the pandemic. Demographic composition of the sample was reviewed regularly in order to inform subsequent invitations to participate in an interview. We approached 25 for interview, four of whom could not be contacted after multiple attempts at different times of day. We obtained full informed consent and carried out interviews with 21 parents between 29 May and 31 July 2020 lasting around 30 minutes each (Range: 18–49 minutes). Demographic information for interviewees is shown in Table 1.

**Table 1. Demographics of interviewees.**

| | Participants (n = 21) | |
|---|---|---|
| | n | % |
| Sex | | |
| Female | 19 | 90.5% |
| Male | 2 | 9.5% |
| Age (years) | | |
| 25–29 | 3 | 14.3% |
| 30–34 | 2 | 9.5% |
| 35–39 | 10 | 47.6% |
| 40–44 | 4 | 19.0% |
| 45–49 | 2 | 9.5% |
| IMD decile* | | |
| 9 and 10 (least deprived) | 7 | 33.3% |
| 7 and 8 | 3 | 14.3% |
| 5 and 6 | 5 | 23.8% |
| 3 and 4 | 2 | 9.5% |
| 1 and 2 (most deprived) | 1 | 4.8% |
| Missing** | 3 | 14.3% |
| Number of children | | |
| 1 | 4 | 19.0% |
| 2 | 11 | 52.4% |
| 3 | 5 | 23.8% |
| 4 | 1 | 4.8% |
| Age of youngest child | | |
| Under 12 months | 3 | 14.3% |
| 1–3 years | 12 | 57.1% |
| 4–6 years | 1 | 4.8% |
| 7–9 years | 5 | 23.8% |
| Used Emergency Department for children in past 12 months | | |
| Yes | 14 | 66.7% |
| No | 7 | 33.3% |
| Used health services during lockdown | | |
| Yes | 12 | 57.1% |
| No | 8 | 38.1% |
| Missing | 1 | 4.8% |
| Decision making affected by COVID-19 | | |
| Yes | 8 | 38.1% |
| No | 11 | 52.4% |
| Missing | 2 | 9.5% |

*While IMD Decile was used for purposive sampling, this is presented as Quintiles for brevity.

**IMD could not be calculated in cases where respondents provided an incomplete postcode.

## Findings

Parents raised a range of issues during interviews, with many describing a spectrum of intellectual and emotional difficulties in interpreting information related to COVID-19, and their responsibilities towards family and wider society. For some parents, this had been a particularly difficult period whilst for others, the situation was less challenging to navigate. We

identified three major themes relating to risk, information, and decision-making, and provide illustrative quotes accompanied by and IMD decile and the age of the participants' children because these are both well known to influence usage of emergency healthcare services [15]. While the motivation for this research had been to understand reasons for the reduction in Emergency Department (ED) use, it became clear from the interviews that parents often referred more widely to services for acute medical conditions which sometimes included healthcare routes such as GPs or walk-in centres.

**Theme 1: Making sense of risk.** Risk was a major topic in these interviews and was discussed at length by parents. This theme describes parents' conceptualisation of risks related to COVID-19 at a point early on in the pandemic and how these could impact decisions to seek care. Parents differentiated risk *to* children from COVID-19 and the potential risk *from* children to the wider community through passing on the virus. The first concerned the direct risk of harm to children from COVID-19; the second concerned the risk that infected children posed to families and society. Parents also discussed these risks changing over the course of the pandemic and how lockdown measures had the potential to modify risks of requiring health services

Most parents appeared to regard the risk of their children being seriously affected through contracting COVID-19 as small; often citing emerging information as underpinning this view:

*I've read quite a bit about risk to children of coronavirus and so whilst I've still got concern around it, my worry isn't particularly that my children will get it and get ill from it.*

(Interview 3: IMD 3, children aged 10 and 7)

Views to the contrary were minimal in our data, although some emphasised that even though emerging risk data around children were reassuring, this didn't stop them worrying. Parents sometimes made reference to the fact that the virus was newly discovered and not fully understood:

*in the back of my mind, it's not well known of what any of this virus is or who it attacks or when, I don't know. It's just like, if I had the personal choice, I wouldn't be exposing myself or my family at any NHS [hospital], yes.*

(Interview 5: IMD 3, child aged 1)

Parents described concerns that hospitals could be a source of infection. For example, being in a confined space, with young children who were not able to physically distance. These perceptions were described with reference to pre-Covid, as well as the use of EDs during the pandemic. A parent who had used the ED during the early stages of the pandemic recounted how they perceived the physical environment as creating a risk of contracting COVID-19:

*So there was a tiny little waiting room and there was four, I think it was four other children, maybe five other children plus one or two parents each in this tiny room. . . I'm like, "Could you please open a window?" "No, we're not allowed, you have to stay in here. No, we're not allowed to open a window." We spent four hours, I think. I realise that you didn't say anything about the time, but we spent four hours in a tiny room with three or four other kids, people coming and going, all potentially with Coronavirus. I think, if we were going to catch it, that was probably our most risky bit of exposure in the waiting room.*

(Interview 2: IMD 9, children aged 8, 4 and 1)

Although the opposite was described by another parent who felt reassured that the physical environment in a hospital created minimal risk of contracting COVID-19:

*I mean I'm reassured now that I've been, and I've seen how well everything was managed and I mean to be honest, the baby has been referred to "Name of hospital 1". So, I'm a lot more relaxed about having to take her since being to"Hospital 2"and"Hospital 3". So, obviously it felt really safe and it felt really clean.*

(Interview 15 (not linked to Interview 2): IMD 10, children aged 8, 4 and 1)

Although these data highlight a range of experiences, they suggest that previous experience of emergency health services as well as those during the pandemic had the potential to reassure or cause concern amongst parents. As time progressed, parents perceived that hospitals were likely to become better able to manage these risks:

*I would not be particularly worried about them catching Covid in paediatric A&E. . .I think that that would probably not be a high risk situation for them. I imagine now that the hospital is so slick*

(Interview 4: IMD 10, children aged 6 and 3)

Although most of the discussion of risk *to* children centred on the potential for them to contract the virus, another issue raised by a parent was that COVID-19 precautions themselves had the potential to be distressing for young children; highlighting the impact of the use of personal protective equipment by clinical staff:

*The biggest impact was on the ward, because"name of son"being three and a half, he's been into hospital before, we had to wait obviously in the bay and he was so, so upset and distressed by the masks.*

(Interview 21: IMD 5, children aged 9, 7 and 3)

Although parents described concerns about their children catching COVID-19 and some worried about the associated risks, these seemed generally to be understood as fairly minimal. On the other hand, parents described worrying about the impact of their child passing on COVID-19 to others.

Parents described concern that their child could pass the virus on to elderly relatives or the wider community:

*It would be foolish for me to say I don't think there's any danger at all [to my child from COVID-19] but, yes, more that you want to prevent the spread as much as possible*

(Interview 10: IMD missing, children aged 10 and 8)

Some parents spoke of concerns that their child could catch COVID-19 in an ED then potentially pass it on to the wider community. One parent described concern about spreading the virus as well as the implications of self-isolating:

*My concerns would be that any of us could catch it, but I'm not really worried about that. I'd just be more worried about spreading it to the outer community and the implications like of everyone having to self-isolate.*

(Interview 7: IMD missing, children aged 4 and 2)

Similarly, a clinical environment could be seen as a potential place where a child could pass the virus to someone vulnerable:

*I'd be reluctant for example to take them into a medical situation in case they were in contact with people who were more vulnerable.*

(Interview 10: IMD missing, children aged 10 and 8)

Although not directly related to risks of the virus, a further point was made by a small number of parents regarding how lockdown measures had the potential to change the likelihood of requiring health services as their children were not in contact with other children and engaging in fewer physical activities with potential to cause injuries:

*So, the fact that they weren't interacting, they weren't picking up these colds, they were healthy overall and definitely led to fewer illnesses. And also, the lockdown in general. . . not the specific nursery, but the fact we weren't going to play parks and running around and stuff as well there was a lot less bruises, cuts, all the things that come with children playing around.*

(Interview 17, IMD missing, children aged 3 and 1)

**Theme 2: Interpreting information on emergency health services.**   This theme focuses on how parents interpreted information about health services during the pandemic. Different types of information were described including government public health advice as well as the internet and social media. Parents also spoke about misinformation about COVID-19 procedures in place in EDs.

Parents described how information, guidance and rules influenced their thoughts and actions on taking their child/children to the ED during this period. Most understood that emergency health services remained available throughout:

*I think there's been enough encouragement that if you've got an unwell child they should be seen. I certainly haven't seen anything to say otherwise.*

(Interview 17, IMD missing, children aged 3 and 1)

However, this was not universal. Several parents interpreted the 'Stay home, Protect the NHS, Save lives' as a clear signal that the NHS should not be used at all:

*'protect the NHS' had that impact, if there's any worries apart from Covid then stay away, quite a blunt message.*

(Interview 3: IMD 3, children aged 10 and 7)

Other parents described the existence of misinformation online, for example one had viewed information stating that children would be separated from parents if they were suspected of having COVID-19 and the potential for this to cause worry.

*There's lots of stuff going round on mums' groups and things online that say, if your child develops Coronavirus symptoms, they will be taken by themselves in an ambulance. . ..and all*

*this stuff which isn't helpful but I don't think that's the case. I don't think hospitals want loads of unaccompanied children loose*

(Interview 2: IMD 9, children aged 8, 4 and 1)

While this participant identified this as likely to be untrue, another made reference to maternity services and how the nature of Covid-related service changes was unclear:

*But there's a lot of fear, I think, amongst pregnant women about kind of how birth works and, obviously, not being able to have your partner in. Or the procedure, whether you would have to wear a mask for it and all that sort of stuff. We were lucky enough to have"name of child"- just before that happened. But kind of, I wanted to send my husband home for the night to sleep after we had the baby, but I said to him I was scared that he wouldn't be allowed to come back in because the situation was changing so rapidly.*

(Interview 14: IMD 6, child aged under 1)

This was echoed by another participant who felt that, although there was good health information available online from the NHS, it wasn't clear how the hospitals were currently functioning under COVID-19 restrictions:

*As I said, the NHS guidance online is really good. If you really want to go and find stuff there's plenty of things available. In terms of what my children are like and how they're doing, I don't feel that there's a lack of information. I just think there's a lack of publicity about what it's like in hospitals at the moment.*

(Interview 10: IMD missing, children aged 10 and 8)

**Theme 3: Making the right decision.**   This theme describes ways in which parents made decisions regarding the use of emergency healthcare for their children. Parents described using their judgement to balance risks and also their desire to behave responsibly and in line with government guidance. Parents who had made healthcare decisions during the pandemic described and reflected on these choices while for others this was a hypothetical scenario.

All of those who had used the ED during the pandemic had been advised to attend after seeking advice from the NHS 111 telephone line. One parent described seeking emergency care due to perceiving that the condition was not something that they could manage themselves:

*If it had been just a finger or a cut or something, I probably would have been able to deal with it myself but with it being a head injury I wanted him checked out properly by a professional to make sure that there was no concussion or anything going on.*

(Interview 13, IMD 6, children aged 9, 5 and under 1)

While it was clear to most parents that emergency services were available if needed, some confusion was sown by lack of access to other types of NHS care, for example a parent of a newborn reported difficulty accessing healthcare professionals which meant that they did not access any services:

*And then with lockdown kind of everybody thought we should be seeing somebody, but nobody wanted it to be them. . .if that makes sense. So, we kind of just muddled through, basically. And still haven't really had him [child] weighed but are just kind of using our own*

*assessment, which still just feels a little bit like... it's our first baby, so not sure if we're doing things right or not essentially.*

(Interview 14: IMD 6, child aged under 1)

Parents described accessing emergency services only when it was deemed absolutely necessary although some highlighted that this approach was no different with their pre-pandemic practice:

*I don't think the criteria would have been very different, I generally would be quite careful about what situations would be appropriate to take them to A&E anyway.*

(Interview 3: IMD 3, children aged 10 and 7)

Indeed, most parents emphasised that they would not avoid seeking emergency care if their child needed it, regardless of the pandemic. One parent described being so focused on getting medical attention for their child that they didn't think about COVID-19:

*I had kind of forgotten about it [COVID-19] and that sounds really strange but I had a son who, for all intents and purposes, had received a bite from a dog, needed to go and get some medical attention and that was it.*

(Interview 18: IMD 5, children aged 5 and 1)

Parents argued that not seeking care for their child was a greater concern than COVID-19.

*I think the risk of not seeking help when you need it is far greater than the risk of being exposed to the Covid virus.*

(Interview 1: IMD 7, children aged 6 and 1)

While for some, decision making was described as being straightforward, other participants referred to situations they had encountered during the pandemic where they found it difficult to decide on the best course of action:

*It was a very sort of, 'do I take him, do I not'...[but] I would never forgive myself if I didn't take him.*

(Interview 13, IMD 6, children aged 9, 5 and under 1)

Several parents said that they avoided seeking healthcare for their children during lockdown. One gave a specific example of delaying seeking care to avoid pressurising healthcare professionals and services:

*That's another thing with my 18 month old, I've actually put off taking her to the GP for quite a while because she's got a limp on one leg and I took her, it will be two weeks on Monday, she went to the GP and I wish I took her sooner but with everything going on, you didn't want to put any extra pressure on the doctors.*

(Interview 15 (not linked to Interview 2): IMD 10, children aged 8, 4 and 1)

Another parent reported deciding against making a follow up appointment after receiving a visit from a paramedic, describing a mixture of concerns about availability of appointments, but also feeling that there were other patients who may be in greater need:

*. . .were aware that the doctors was probably, massively overly used at that point because of all this Covid so we were like we're not going to get an appointment or, even if we do, there's probably somebody who needs it more than us.*

(Interview 16: IMD 10, children aged 4 and 1)

Others echoed this sentiment of increased caution in accessing emergency care; one parent described how the pandemic had impacted their decision making criteria in terms of 'raising the bar" for when to seek care, also describing how they would take efforts to self-manage at home:

*That's the big thing, trying to do what you can at home, hoping that we're putting less pressure on the service and then yes, just raising the bar. Do what we can at home so we're less likely to, A, infect ourselves from being there and B, put pressure on the service, yes. Definitely raising the bar.*

(Interview 2: IMD 9, children aged 8, 4 and 1)

Similarly, one parent whose child had an existing medical condition and was very accustomed to attending the ED, described trying to find alternatives to avoid having to attend:

*I tried ringing staff earlier on in the day, nobody got back to me because obviously there's a lot going on and I haven't kept on. So, normally I would have just gone straight in, gone into A&E and I just didn't't in this instance. I just kept ringing round and round and round until I got hold of a consultant to talk to somebody.*

(Interview 6: IMD 5, children aged 13 and 9)

Several parents emphasised the value of accessing health advice prior to attending in-person emergency services, frequently citing NHS 111 but also GPs or opinions of friends in healthcare roles. This was important in providing reassurance that parents were taking the correct course of action:

*Yes, so I used 111 when obviously"name of son"was poorly and I have used it in the past but you know what, I think the times I've used it recently has always been about his viral wheeze, when generally you're not quite happy with his breathing, you want that advice and then they say, actually yes what you're telling me, you probably need to come in. But both instances have been when the GP has not been open, I would probably go to the GP first if they were open and if they weren't, then obviously I wouldn't hesitate to call 111.*

(Interview 21: IMD 5, children aged 9, 7 and 3)

While no one in our sample described being completely unwilling to using emergency services, one participant described a friend who decided against taking their child for emergency treatment because of COVID-19 concerns:

*A friend's child slipped in the bath, he cut his chin open and there as blood all over. She said, "I'm not taking him to the hospital, the risk of Coronavirus, I'm not taking him." So she cleaned his face up and then just got some Elastoplast, whatever, held it together and just made her own little stitchy, sticky things to close the gap.*

(Interview 2: IMD 9, children aged 8, 4 and 1)

## Discussion

Our data provide a snapshot of views of a group of parents during the first wave of the COVID-19 pandemic in the UK. This research is part of a growing body of literature which describes short and longer-term concerns about the impact of COVID-19 on the health and wellbeing of children [16–21].

While the rapidly changing landscape with regard to government policy and messaging provided a challenging context in which to interpret and report such findings, we believe they are important in contributing towards an understanding of parents' views on accessing emergency healthcare at the beginning of a pandemic and how these views may be impacted by government policy, public health information and messaging. They are also important in terms of ensuring that the learning from the Covid pandemic is used to inform future emergency planning, and access to emergency healthcare in normal times.

We have described how parents weighed up competing information sources, perceptions of risk to themselves and to their children, government guidelines and rules, and a desire to reduce burden on the NHS, when considering whether to seek emergency healthcare services for their children during the COVID-19 pandemic. Whilst parents usually said they would seek care if it was needed, there were examples where this was not the case. Parents were less concerned that their child might become ill from hospital-acquired COVID-19 than their potential to become infected and then infect others. There was considerable divergence in our sample regarding whether the "Stay at Home" and "Protect the NHS" government messaging encompassed seeking emergency healthcare.

While the context in which parents were making decisions was clearly very different to pre-pandemic times, and several studies suggest that attendance at pediatric emergency settings dropped dramatically, it is still important to consider whether decision making processes themselves have changed. With reference to our themes, forms of 'Making sense of Risk' have been seen in other studies, in which parents make decisions based on the perceived seriousness of children's' health condition prior to seeking care [9, 10, 22], although in these cases 'risk' related to the danger to the child from the health condition, rather than the risk to children and families (and the wider community) as a consequence of seeking care. Similarly, elements of our theme 'Interpreting information on emergency health services' have been seen in previous studies in which parents have made decisions to use emergency services depending on the availability of services at particular times [9]. While availability of services may be a longstanding consideration, the pandemic context saw additional public health messaging relating to protecting health services, as well as perceptions on if and how services were functioning with the added pressures of COVID-19. Our final theme, 'Making the right decision' captured elements of previously described behavior in which parents balance competing factors before whether deciding whether or not to access health services [9], this theme was of particular interest in relation to questions of whether some parents may delay accessing healthcare for their children; while parents generally felt that they would always seek care where there was an urgent need, some described using emergency services although conversely some described delaying seeking healthcare. Others described raising of the bar; being more discerning about when they would access healthcare, being more likely to self-manage at home, this finding echoes the concept of a 'theshold' described previously [10]. These added factors due to COVID-19 appeared to make an already difficult decision making process [23] even more complex.

Hesitancy in seeking healthcare during COVID-19 has been previously described in children with cancer [24] and in the general population. In a survey of 1,044 parents in Ireland, 22% of those identifying a need for emergency healthcare for their children did not seek it [25]. Others have sought to understand the views of parents accessing vaccination for young

children, with parents needing to weigh up multiple competing factors [26], similar to the process described by parents in our study.

Our findings provide some insight into parental attitudes towards the use of emergency healthcare during the COVID-19 pandemic and complement British Paediatric Surveillance Unit data describing perceptions of delayed-presentation to hospital among British paediatricians [5]. A French study published before the pandemic described six motives for parents attending the ED, finding that parents were serious and goal-orientated in their seeking of this healthcare [27]. Despite arguments to the contrary from professionals, parents would be unlikely to agree that visits for apparent minor-illness were 'inappropriate' [27]. Our findings are in accordance; parents took their decision of whether and how to seek healthcare incredibly seriously and described ways in which they tried to reduce healthcare seeking during the pandemic. A questionnaire-based project in Ireland attempted to answer similar questions to our study; in addition to our shared findings of misinterpretation of the 'Stay at home' messaging and perception of risk, this study reported the importance of parental stress, measured using the stress subscale of the Depression Anxiety Stress Scale [28].

Our study adds to this body of literature by describing the views of parents directly and exploring decision-making around seeking urgent care at the beginning of the COVID-19 pandemic. Interviews were conducted via telephone, minimising participant inconvenience and maintaining physical distancing. One potential limitation is that we did not interview children, who are often active partners in making some of these decisions. The study population was relatively small, and represents those with motivation to participate, rather than the whole population. The sample was also skewed to a less deprived demographic than the general population. It is known that emergency service use by children and young people is higher in populations experiencing higher levels of deprivation [9, 29] therefore this is a limitation of our study. Respondent validation methods may have been helpful in ensuring that our interpretation of data were correct from a participant perspective.

A Canadian paper made recommendations for minimising indirect effects of COVID-19 on children and young people's health which included developing responsive health systems, communicate with children and families and use data to inform decision making [6]. Our findings similarly emphasise the need for clear communication and have several implications for providers of paediatric emergency healthcare services, both during the current pandemic and in future emergency planning in times of crisis. First, providers should recognise the complex decision-making processes that parents perform in determining whether and when to seek help. As discussed, this process was likely to have been strongly influenced by national COVID-19 messaging. Local providers are well placed to understand specific needs of their communities, and provide crucial local context to broad national messaging; these providers could consider developing media-messaging, for example via social-media platforms, targeted locally to reassure parents. We have discussed the internal 'threshold' that parents described having to reach in order to seek care during the pandemic. This metaphor might also be useful for organisations attempting to reduce emergency attendances in other contexts, for example by redirecting to other elements of the healthcare system.

Future research should attempt to identify specific groups who are less likely to attend when required, so that strategies to optimise use of emergency healthcare services, whilst avoiding harm, can be implemented.

"Stay at home" messages have been used at several points during this pandemic, and may be used again in future disease outbreaks. This research suggests that this messaging may have led to adverse consequences, the impacts of which require further study. At the very least, we suggest that such messaging should be accompanied by clear statements that seeking emergency healthcare should always be prioritised. In a post-pandemic landscape, this study may

be viewed as part of a wider body of literature concerned with longstanding issues of supporting families to make decisions regarding appropriate use of health services.

## Supporting information

**S1 File.**
(DOCX)

## Acknowledgments

We would like to thank all of the participants who took the time to speak to us and made this research possible.

## Author Contributions

**Conceptualization:** Matthew Breckons, Sophie Thorne, Rebecca Walsh, Sunil Bhopal, Stephen Owens, Judith Rankin.

**Data curation:** Matthew Breckons, Sophie Thorne, Rebecca Walsh.

**Formal analysis:** Matthew Breckons, Sophie Thorne, Rebecca Walsh, Sunil Bhopal, Stephen Owens, Judith Rankin.

**Methodology:** Matthew Breckons, Sophie Thorne, Rebecca Walsh, Sunil Bhopal, Stephen Owens, Judith Rankin.

**Project administration:** Matthew Breckons.

**Writing – original draft:** Matthew Breckons, Sunil Bhopal, Stephen Owens, Judith Rankin.

**Writing – review & editing:** Matthew Breckons, Sophie Thorne, Rebecca Walsh, Sunil Bhopal, Stephen Owens, Judith Rankin.

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
