## [Decision Letter · Decision Letter 0]

10 Mar 2022

PONE-D-21-33201Why did the children stop coming? Reasons for paediatric emergency department attendance decrease during the first wave of the COVID-19 pandemic in the United Kingdom: A qualitative studyPLOS ONE

Dear Dr. Breckons,

Thank you for submitting your manuscript to PLOS ONE. After careful consideration, we feel that it has merit but does not fully meet PLOS ONE’s publication criteria as it currently stands. Therefore, we invite you to submit a revised version of the manuscript that addresses the points raised during the review process.

In particular, I would urge you to consider and resolve the methodological issues raised by the reviewer.

We look forward to receiving your revised manuscript.

Kind regards,

Enrique Castro-Sánchez

Academic Editor

PLOS ONE

Journal Requirements:

Reviewers' comments:

Reviewer's Responses to Questions

**Comments to the Author**

1. Is the manuscript technically sound, and do the data support the conclusions?

Reviewer #1: No

2. Has the statistical analysis been performed appropriately and rigorously? 

Reviewer #1: Yes

3. Have the authors made all data underlying the findings in their manuscript fully available?

Reviewer #1: No

4. Is the manuscript presented in an intelligible fashion and written in standard English?

Reviewer #1: Yes

5. Review Comments to the Author

Reviewer #1: This study addresses an important concern raised by healthcare professionals i.e. the drop in paediatric attendance at health facilities during the COVID-19 pandemic. The topic has received much attention in the literature with several studies published over the past 18 months.

This manuscript describes a qualitative study aimed at understanding parents ’experiences and views on the use of children’s urgent healthcare services during the first wave of the Covid-19 pandemic. The authors state their objectives were: to understand if and how urgent healthcare use, self-care and information needs differed during ‘lockdown ’and to consider parent perceptions of risks of Covid-19. This seems somewhat inconsistent with the title of the paper “Why did the children stop coming? Reasons for paediatric emergency department attendance decrease during the first wave of the COVID-19 pandemic in the United Kingdom: A qualitative study”. There are several problems in the study design that create challenges to achieving the objectives or indeed addressing the question posed in the title of the paper.

Firstly there is the question of how healthcare use and information needs differed during the pandemic. To adequately address this requires some comparison with the situation in non-pandemic times. See for example https://onlinelibrary.wiley.com/doi/full/10.1111/hex.13305 Qualitative study of parental health-seeking behaviour published in July 2021.

Another challenge referred to in the manuscript "Accessing views of people who are not using a service is, by its nature, challenging; we therefore sought to gain views from a more general sample of parents on using urgent healthcare during the pandemic, being aware that this sample was likely to include people who had and had not needed to make urgent healthcare decisions during the pandemic" is one that impacts significantly on the relevance of the findings to the question posed at the outset. The sample is heavily skewed towards those who used the emergency department services (14 had used and 7 had not) and as such are not in the best position to answer the question "Why did the children stop coming?"

A further limitation in the sample is the high proportion with IMD decile at 5 or above (15) compared to those below 5 (3), as the authors state that IMD is a variable that impacts use of emergency services.

Both of these are significant limitations that may be substantially skewing the results and as such should be acknowledged and discussed in the paper.

There are some other elements of the study design that need explanation:

1. What was that rationale of the focus on 0-8 year olds?

2. Why does the table then include parents of children up to 9 years of age?

3. What informed the interview guide? No evidence is presented of a literature review on parental decision making in accessing emergency care?

4. Interview guide should be included as an appendix to the paper.

5. How was the deductive element of the analysis framework constructed? Did the authors carry out any exploratory work with parents? Was it informed by the literature?

This paper https://bmchealthservres.biomedcentral.com/articles/10.1186/s12913-020-05527-5 on Factors that influence family and parental preferences and decision making for unscheduled paediatric healthcare – systematic review published in July 2020 is particularly relevant. Although data collection for the study occurred before the publication of this paper, the protocol for the review was published in 2019 https://doi.org/10.12688/hrbopenres.12897.2.

Discussion section - I feel this section needs further work to demonstrate the contribution of the study to the growing body of literature in this area. Through the authors refer to some recent studies in the area, they have not adequately situated their study findings in relation to the findings of previous studies. They have also omitted from the discussion any reference to the policy implications of Chanchlani et al paper (mentioned only briefly in the introduction) https://www.cmaj.ca/content/192/32/E921.short Addressing the indirect effects of COVID-19 on the health of children and young people published in August 2020, and they have omitted to reference an equally relevant paper by McDonnell et al https://www.mdpi.com/1660-4601/17/18/6719/htm Assessing the Impact of COVID-19 Public Health Stages on Paediatric Emergency Attendance published in September 2020.

I disagree with the authors claim in the discussion “Our findings provide explanations for some of the large reduction in emergency healthcare utilisation seen in EDs” owing to the skewed nature of the sample as mentioned earlier.

In summary, this paper despite addressing an important topic, needs substantial revision to demonstrate its contribution to the literature in this area, particularly given the significant limitations in study design and sample choice.

6. PLOS authors have the option to publish the peer review history of their article (what does this mean?). If published, this will include your full peer review and any attached files.

Reviewer #1: **Yes: **Eilish McAuliffe

---

## [Author Response · Author response to Decision Letter 0]

12 Sep 2022

Dear Reviewers,

Thank you for your helpful feedback:

Your comments were extremely helpful, particularly in bringing our work into clearer focus with regards to existing literature about parental decision making about their children’s healthcare. While we collected data at the start of the Covid-19 pandemic and much has changed in this period, we still believe that our work contributes to understanding about the factors which parents and carers weigh up when considering use of healthcare for children. I hope that we have adequately addressed all review feedback but please let us know if you have any further queries.

Please find our changes detailed below next to each of your comments

1) The authors state their objectives were: to understand if and how urgent healthcare use, self-care and information needs differed during ‘lockdown ’and to consider parent perceptions of risks of Covid-19. This seems somewhat inconsistent with the title of the paper “Why did the children stop coming? Reasons for paediatric emergency department attendance decrease during the first wave of the COVID-19 pandemic in the United Kingdom: A qualitative study”.

RESPONSE: This is a helpful point. We originally set out with a very pragmatic and urgent research question grounded in experiences of clinical members of the research team of reduced attendance in the local ED. However we appreciate that we did not conclusively answer the question that our title poses, and that our methods (in particular our recruitment) limited our ability to do so. We do, however, believe that our work adds a valuable perspective on access to childrens’ healthcare services during the pandemic. We have amended our title to: Parental perspectives on emergency health service use during the first wave of the COVID-19 pandemic in the United Kingdom: A qualitative study which we hope you agree is an appropriate title. We have also amended our stated aims in line with this to emphasise that we sought to understand views and decision making. 

2) There are several problems in the study design that create challenges to achieving the objectives or indeed addressing the question posed in the title of the paper.

Firstly there is the question of how healthcare use and information needs differed during the pandemic. To adequately address this requires some comparison with the situation in non-pandemic times. See for example https://onlinelibrary.wiley.com/doi/full/10.1111/hex.13305 Qualitative study of parental health-seeking behaviour published in July 2021.

RESPONSE: Thank you for this comment. We have now added extensive discussion of the themes we describe in relation to existing literature to emphasise that there are elements of our themes contained in previous work but that Covid-19 has added elements to many of these. 

3) Another challenge referred to in the manuscript "Accessing views of people who are not using a service is, by its nature, challenging; we therefore sought to gain views from a more general sample of parents on using urgent healthcare during the pandemic, being aware that this sample was likely to include people who had and had not needed to make urgent healthcare decisions during the pandemic" is one that impacts significantly on the relevance of the findings to the question posed at the outset.

The sample is heavily skewed towards those who used the emergency department services (14 had used and 7 had not) and as such are not in the best position to answer the question "Why did the children stop coming?"

RESPONSE: We agree with this assertion, we hope that the change in title and the rewording of our aims and additional discussion of this point have helped to provide some clarity on this. 

4) A further limitation in the sample is the high proportion with IMD decile at 5 or above (15) compared to those below 5 (3), as the authors state that IMD is a variable that impacts use of emergency services.

RESPONSE: We have cited this in the discussion but elaborated further why this is an important limitation.

5) What was that rationale of the focus on 0-8 year olds?

RESPONSE: Apologies this was a typographic error. We debated the cut off at length within the study team and in the end went with the inclusion criteria of ‘parents of children up to 10 years old’. While we appreciate we could have cut off our sample in various ways (for example several studies have defined young children as 0-8 and in the reference you provided the researchers used 0-12) this focus reflected inclusion of ‘pre-school children’ and ‘school children’ in a large UK survey of healthcare use in children under 15 years: (Ruzangi, J., Blair, M., Cecil, E., Greenfield, G., Bottle, A., Hargreaves, D.S. and Saxena, S., 2020. Trends in healthcare use in children aged less than 15 years: a population-based cohort study in England from 2007 to 2017. BMJ open, 10(5), p.e033761.) We excluded the 10-15year old group who this study defined as teenagers. 

6) Why does the table then include parents of children up to 9 years of age?

RESPONSE: As above – apologies for this error .

7) What informed the interview guide? No evidence is presented of a literature review on parental decision making in accessing emergency care?

RESPONSE: Our topic guide included a small number of questions to probe participants’ views on seeking healthcare during the pandemic and if they felt there were differences to their pre-pandemic views. 

8) Interview guide should be included as an appendix to the paper.

RESPONSE: We have added this as requested.

9) How was the deductive element of the analysis framework constructed? Did the authors carry out any exploratory work with parents? Was it informed by the literature?

RESPONSE: On reflection it was confusing to describe a deductive approach and we have tried to clarify this in our methods. We did not carry out exploratory work with parents. 

10) This paper https://bmchealthservres.biomedcentral.com/articles/10.1186/s12913-020-05527-5 on Factors that influence family and parental preferences and decision making for unscheduled paediatric healthcare – systematic review published in July 2020 is particularly relevant. Although data collection for the study occurred before the publication of this paper, the protocol for the review was published in 2019 https://doi.org/10.12688/hrbopenres.12897.2.

RESPONSE: Thank you for this helpful suggestion. We have incorporated this into our discussion. 

11) Discussion section - I feel this section needs further work to demonstrate the contribution of the study to the growing body of literature in this area. Through the authors refer to some recent studies in the area, they have not adequately situated their study findings in relation to the findings of previous studies. They have also omitted from the discussion any reference to the policy implications of Chanchlani et al paper (mentioned only briefly in the introduction) https://www.cmaj.ca/content/192/32/E921.short Addressing the indirect effects of COVID-19 on the health of children and young people published in August 2020, and they have omitted to reference an equally relevant paper by McDonnell et al https://www.mdpi.com/1660-4601/17/18/6719/htm Assessing the Impact of COVID-19 Public Health Stages on Paediatric Emergency Attendance published in September 2020.

RESPONSE: Thank you for this comment. This is helpful in situating our paper within this topic area. We have now made reference to both of these papers in our manuscript. 

12) I disagree with the authors claim in the discussion “Our findings provide explanations for some of the large reduction in emergency healthcare utilisation seen in EDs” owing to the skewed nature of the sample as mentioned earlier.

RESPONSE: Thank you for this comment. We have reframed our discussion to suggest that our study offers insight into attitudes into the use of emergency healthcare during the Covid-19 pandemic. We hope that this is a more appropriate claim. 

13) In summary, this paper despite addressing an important topic, needs substantial revision to demonstrate its contribution to the literature in this area, particularly given the significant limitations in study design and sample choice.

RESPONSE: We hope that our changes described above have addressed these concerns and queries.

---

## [Decision Letter · Decision Letter 1]

13 Feb 2023

PONE-D-21-33201R1Parental perspectives on emergency health service use during the first wave of the COVID-19 pandemic in the United Kingdom: A qualitative studyPLOS ONE

Dear Matthew,

Thank you for submitting your manuscript to PLOS ONE. After careful consideration, we feel that it has merit but does not fully meet PLOS ONE’s publication criteria as it currently stands. Therefore, we invite you to submit a revised version of the manuscript that addresses the points raised during the review process.

Please ensure you address the reviewer's comments in your next revised version. These are on the whole minor amendments and we note that this is an improved version.

We look forward to receiving your revised manuscript.

Kind regards,

Cari Malcolm

Academic Editor

PLOS ONE

Journal Requirements:

Reviewers' comments:

Reviewer's Responses to Questions

**Comments to the Author**

1. If the authors have adequately addressed your comments raised in a previous round of review and you feel that this manuscript is now acceptable for publication, you may indicate that here to bypass the “Comments to the Author” section, enter your conflict of interest statement in the “Confidential to Editor” section, and submit your "Accept" recommendation.

Reviewer #2: (No Response)

Reviewer #3: All comments have been addressed

2. Is the manuscript technically sound, and do the data support the conclusions?

Reviewer #2: Partly

Reviewer #3: Yes

3. Has the statistical analysis been performed appropriately and rigorously? 

Reviewer #2: N/A

Reviewer #3: N/A

4. Have the authors made all data underlying the findings in their manuscript fully available?

Reviewer #2: Yes

Reviewer #3: No

5. Is the manuscript presented in an intelligible fashion and written in standard English?

Reviewer #2: Yes

Reviewer #3: Yes

6. Review Comments to the Author

Reviewer #2: Thank you for this important manuscript on children's access to emergency healthcare during the early part of the COVID-19 pandemic.

In summary, parents of children were recruited for interviews via social media and posters to discuss their decision making around accessing emergency healthcare services during the early COVID-19 pandemic. A skewed population was recruited; over 2/3 of participants had accessed healthcare services during the pandemic and the majority originate from areas of low deprivation.

Three themes were identified:

1) Making sense of risk. Whilst this is a key part of the paper, the findings mainly relate to the risk that interviewees felt at the time of accessing healthcare settings, rather than the anticipated risks which influenced their decision making around accessing healthcare settings.

2) Interpreting information on emergency healthcare services. This is a really key part of the paper. It is unfortunate that the thematic questioning did not include the use of online resources - NHS, RCPCH etc all released early guidance and understanding whether the use of these resources influenced decision making for families would be an important message to draw out for any future pandemics

3) Making the right decision. There appears to be a well-rounded description of view points within this theme.

The authors provide a balanced discussion and reach reasonable conclusions on the data available from this study and gives an additional important parental voice to the literature on this.

Reviewer #3: I have had the opportunity to review the manuscript. First off, I would like to apologise for my delay in reviewing. As I was not one of the original reviewers I have had to familiarise myself with the manuscript, the reviewers commentary and your responses, and the revised manuscript. Having had this opportunity I offer the following commentary, however, I have attempted to keep it light-touch as I realise this is the second round of reviews:

GENERAL:

- Could you please capitalise COVID-19 and ensure this is used throughout.

- I have suggested a minor revision, but will make it clear to the Editor that as long as these changes are made, I do not need to see the paper back, nor do I wish to see you subjected to another round of reviews.

INTRODUCTION:

- You may wish to explain that social distancing is a bit of a misnomer, when in fact the reality was physical distancing and we were encouraged to socially network.

- A personal point of pedantry - line 54 of the revised introduction, can you re-phrase the beginning of the sentence so it doesn't refer to 'A 2020 paper' - when there are no written references with this style it adds a mystery to the tone which is confusing - perhaps name the authors?

METHODS:

- Interview schedule not topic guide, the latter being reserved for unstructured interviews and focus groups - change throughout.

- You state you recruited on IMD Decile, but you report IMD grouped as Quintiles - perhaps explain why?

- Could a note be put in as to why some IMD data is missing?

RESULTS:

- Lines 183 - 197 in revised results section - is it sheer coincidence that the participants in Interview 2 and Interview 15 have children of the exact same age? At first I thought this was two parents of the same children, and then I noticed that they were in different IMD groups, so I perhaps mistakenly assumed they were separated parents of the same children. I think some clarification might be needed here if, in fact, these two participants bear no relationship to one another.

- Where you are redacting names in the results could you utilise the accepted '<name of="" son="">' rather than *Name of Son* - change throughout.

DISCUSSION:

- Is a bit long for my liking, but I can see a lot of it comes from the edits you have had to make for the previous reviewer.</name>

7. PLOS authors have the option to publish the peer review history of their article (what does this mean?). If published, this will include your full peer review and any attached files.

Reviewer #2: No

Reviewer #3: No

---

## [Author Response · Author response to Decision Letter 1]

12 Apr 2023

Dear Editor,

Thank you very much for the chance to amend our manuscript in line with the feedback received from the reviewers. We note that Reviewer #2 did not request changes but provided some general thoughts about our research but where amendments were requested by Reviewer #3 we have made changes. 

Please find these changes detailed in the table below. We note that Reviewer #3 suggested that further review may not be necessary. We look forward to hearing from you.

Yours sincerely, 

Matt Breckons

Reviewer #2 Comment:

In summary, parents of children were recruited for interviews via social media and posters to discuss their decision making around accessing emergency healthcare services during the early COVID-19 pandemic. A skewed population was recruited; over 2/3 of participants had accessed healthcare services during the pandemic and the majority originate from areas of low deprivation.

Three themes were identified:

1) Making sense of risk. Whilst this is a key part of the paper, the findings mainly relate to the risk that interviewees felt at the time of accessing healthcare settings, rather than the anticipated risks which influenced their decision making around accessing healthcare settings.

2) Interpreting information on emergency healthcare services. This is a really key part of the paper. It is unfortunate that the thematic questioning did not include the use of online resources - NHS, RCPCH etc all released early guidance and understanding whether the use of these resources influenced decision making for families would be an important message to draw out for any future pandemics

3) Making the right decision. There appears to be a well-rounded description of view points within this theme.

Our Response:

Thank you for these comments. Your thoughts on parents’ views on risk are interesting; we anticipated that people would talk about perceived risk of attending healthcare settings but as you refer to, the fact that 2/3 had accessed healthcare services meant that perhaps a broader discussion of risk took place. 

With regard to interpreting information on healthcare services. We agree that this is a key part of the paper. While we did not specifically mention these sources of information, it is interesting that these were not specifically mentioned; it was the case that participants discussed sources of information they used which included online information although we agree that a more ‘zoomed in’ discussion of information sources would be interesting.

Your comments re: skewed sample are interesting and does raise the question of how a less skewed population might be accessed – we will certainly take this learning forward for future work.

Reviewer #3 Comment:

Could you please capitalise COVID-19 and ensure this is used throughout.

Our Response:

Thank you, we have made this change throughout.

Reviewer #3 Comment:

You may wish to explain that social distancing is a bit of a misnomer, when in fact the reality was physical distancing and we were encouraged to socially network.

Our Response:

Thank you for this comment. We appreciate the distinction being made more recently. We have considered how we have used this term and agree that ‘physical distancing’ is what we’re referring to in most cases and have revised accordingly.

Reviewer #3 Comment:

A personal point of pedantry - line 54 of the revised introduction, can you re-phrase the beginning of the sentence so it doesn't refer to 'A 2020 paper' - when there are no written references with this style it adds a mystery to the tone which is confusing - perhaps name the authors?

Our Response:

Thank you we have amended this accordingly.

Reviewer #3 Comment:

Interview schedule not topic guide, the latter being reserved for unstructured interviews and focus groups - change throughout.

Our Response:

Thank you we have amended.

Reviewer #3 Comment:

You state you recruited on IMD Decile, but you report IMD grouped as Quintiles - perhaps explain why?

Our Response:

Thank you this is a great point. We have added a footnote to the table to clarify.

Reviewer #3 Comment:

Could a note be put in as to why some IMD data is missing?

Our Response:

We have added a note to clarify that IMD could not be calculated where an incomplete postcode was provided. 

Reviewer #3 Comment:

Lines 183 - 197 in revised results section - is it sheer coincidence that the participants in Interview 2 and Interview 15 have children of the exact same age? At first I thought this was two parents of the same children, and then I noticed that they were in different IMD groups, so I perhaps mistakenly assumed they were separated parents of the same children. I think some clarification might be needed here if, in fact, these two participants bear no relationship to one another.

Our Response:

Yes! We checked meticulously prior to submission but have just returned to these data to check. We appreciate this is coincidence in a small sample and, as such, have added clarification on the two occasions where we refer to Interview 15 – mentioning that there is no link to Interview 2.

Reviewer #3 Comment:

Where you are redacting names in the results could you utilise the accepted '' rather than *Name of Son* - change throughout.

Our Response:

We have changed throughout.

Reviewer #3 Comment:

Is a bit long for my liking, but I can see a lot of it comes from the edits you have had to make for the previous reviewer

Our Response:

Thank you. We agree with this comment.

---

## [Editor Report · Decision Letter 2]

24 Apr 2023

Parental perspectives on emergency health service use during the first wave of the COVID-19 pandemic in the United Kingdom: A qualitative study

PONE-D-21-33201R2

Dear Dr Breckons,

We’re pleased to inform you that your manuscript has been judged scientifically suitable for publication and will be formally accepted for publication once it meets all outstanding technical requirements.

Kind regards,

Cari Malcolm

Academic Editor

PLOS ONE
---

## [Editor Report · Acceptance letter]

4 May 2023

PONE-D-21-33201R2 

Parental perspectives on emergency health service use during the first wave of the COVID-19 pandemic in the United Kingdom: A qualitative study 

Dear Dr. Breckons:

I'm pleased to inform you that your manuscript has been deemed suitable for publication in PLOS ONE. Congratulations! Your manuscript is now with our production department. 

Kind regards, 

on behalf of

Dr. Cari Malcolm 

Academic Editor

PLOS ONE